# Transcriptome Analysis of the Adipose Tissue of Luchuan and Duroc Pigs

**DOI:** 10.3390/ani12172258

**Published:** 2022-08-31

**Authors:** Hongyuan Pan, Tengda Huang, Lin Yu, Peng Wang, Songtao Su, Tian Wu, Yin Bai, Yonghao Teng, Yutian Wei, Lei Zhou, Yixing Li

**Affiliations:** State Key Laboratory for Conservation and Utilization of Subtropical Agro-Bioresources, College of Animal Science and Technology, Guangxi University, Nanning 530004, China

**Keywords:** meat quality, lipid metabolism, fat deposition, transcriptome, Luchuan, Duroc

## Abstract

**Simple Summary:**

Fat is a vital body tissue of pigs and a crucial index that affects the production efficiency of pigs. In this study, Duroc pigs and Luchuan pigs were used as animal materials, transcriptome sequencing was used to compare the back adipose tissue of the two breeds, to explore the key reason of difference in fat deposition. The result provided new ideas and reference for further study of fat development.

**Abstract:**

Fat deposition is a crucial element in pig production that affects production efficiency, quality and consumer choices. In this study, Duroc pigs, a Western, famous lean pig breed, and Luchuan pigs, a Chinese, native obese pig breed, were used as animal materials. Transcriptome sequencing was used to compare the back adipose tissue of Duroc and Luchuan pigs, to explore the key genes regulating fat deposition. The results showed that 418 genes were highly expressed in the Duroc pig, and 441 genes were highly expressed in the Luchuan pig. In addition, the function enrichment analysis disclosed that the DEGs had been primarily enriched in lipid metabolism, storage and transport pathways. Furthermore, significant differences in the metabolic pathways of alpha-linolenic acid, linoleic acid and arachidonic acid explained the differences in the flavor of the two kinds of pork. Finally, the gene set enrichment analysis (GSEA) exposed that the difference in fat deposition between Duroc and Luchuan pigs may be due to the differential regulation of the metabolism pathway of fatty acid. Therefore, this study described the differential expression transcriptional map of adipose tissue of Duroc pig and Luchuan pig, identified the functional genes regulating pig fat deposition, and provided new hypotheses and references for further study of fat development.

## 1. Introduction

Adipose tissue is the main organ to maintain the balance of energy metabolism in animals [1]. Fat is an important economic trait that affects the production efficiency and reproductive performance of pigs [2]. According to its location, adipose tissue can be divided into subcutaneous fat (SCF) and intramuscular fat (IMF), and the body fat is mainly deposited in SCF [3]. Fat deposition is characterized by an increase in the number and size of adipocytes [4]. Fat deposition form and amount are important indicators to evaluate animal meat quality. With the continuous improvement of living standards, peoples’ requirements for meat products have changed from quantity to quality, and people prefer lean meat products [5]. In order to meet peoples’ demands for lean meat products, animal husbandry needs to reduce the content of SCF and improve the lean meat rate of pigs [6].

Duroc pig is one of the major commercial lean pig breeds originating in the United States. It is famous for its thin backfat, high rate of lean meat and fast growth [7]. Luchuan pig is a typical fat species in China. It has thick backfat, high fat content and slow growth [8]. Due to the significant differences in backfat thickness and fat deposition capacity [9], Duroc and Luchuan pigs are ideal models for studying the molecular mechanism of fat deposition.

Transcriptome sequencing can identify and quantify the global changes at the transcriptional level, which is a typical bioinformatics analysis technique and has been widely used in animal husbandry [10]. Shi et al. found that lncRNAs may be involved in regulating subcutaneous fat development [11]. R et al. revealed a new aspect of genetic regulation of fat deposition in pigs [12]. Xing et al. used transcriptome sequencing to provide important insights into miRNA expression patterns in porcine dorsal adipose tissue [13]. Based on weighted gene co-expression network analysis between Landrace and Songliao pigs, Xing et al. identified the key genes affecting fat deposition in pigs [14]. However, the transcriptome study on the comparison of backfat between Duroc and Luchuan pigs has not been conducted yet.

In this study, transcriptome sequencing was performed on backfat tissues of Duroc and Luchuan pigs to explore key genes regulating adipose deposition. This study is of great significance for improving the quality of meat products and provides reference for exploring the regulation mechanism of fat deposition.

## 2. Materials and Methods

### 2.1. Animals and Diet

Duroc boars (*n* = 6) were bought from Guangxi State Farms Yongxin Animal Husbandry Group Co., Ltd. Luchuan boars (*n* = 6) were bought from The Animal Husbandry Research Institute of Guangxi Zhuang Autonomous Region. All pigs had free access to food and water and were fed under the same feeding conditions until 6 months of age. Dietary formula is shown in Table 1 [15]. All pigs were slaughtered at 180 days of age. Backfat adipose tissues were transferred to liquid nitrogen immediately and then stored at −80 °C.

### 2.2. Hematoxylin and Eosin (HE) Staining

Adipose tissues of Duroc and Luchuan pigs were fixed with a 10% buffered formaldehyde solution, followed by paraffin-embedded sections and hematoxylin–eosin staining according to standard procedures. Histological examination was performed under light microscope, followed by morphological analysis using image analysis software Image-Pro Plus 6.0 [9]. At least 20 adipocytes were randomly selected, and the diameter and area of adipocytes in the collected images were measured.

### 2.3. Transcriptome Sequencing

In the transcriptome, we selected 3 adipose tissue samples whose backfat thickness was closest to the mean in each group for high-throughput sequencing. Transcriptome sequencing of backfat adipose tissue was performed by Annoroad Gene Technology Co., Ltd. (Beijing, China). A total of 1.5 ug RNA per sample were individually obtained using TRIzol reagent. Sample purity of RNA was determined using NanoPhotometer^®^ spectrophotometer (IMPLEN, Westlake Village, CA, USA), the samples with OD260/DO280 values (ratio) between 1.8 and 2.0 are considered qualified. The concentration of the samples was tested by Qubit^®^3.0 Fluorometer (Life Technologies, Carlsbad, CA, USA). Total RNA integrity and concentration were measured using Agilent 2100 RNA Nano 6000 Assay Kit (Agilent Technologies, Santa Clara, CA, USA). After the total RNA samples were up to standard, Oligo (dT) beads were used for purification. The fragmentation buffer was used to fragment purified mRNA fragments into short fragments. Using the cDNA fragment as a template, the first strand of cDNA was synthesized with Random Primer 6. The second strand of cDNA was synthesized by adding buffer solution, dNTPs, RnaseH and DNA Polymerase I. QIAquick PCR kit was used for purification and eluted with EB buffer. Finally, end repair, dA-tailing, adapter ligation, and PCR enrichment were performed to complete the construction of the library.

After library construction, Qubit3.0 was used for preliminary quantification. Then, diluted to 1 ng/uL. Agilent 2100 was then used to detect the insert size of the library. After determining the insertion size, Q-PCR was performed using Bio-RAD CFX 96 fluorescence quantitative PCR and Bio-RAD KIT iQ SYBR GRN. The effective concentration of the library was quantified accurately (effective concentration > 10 nM). The qualified libraries were sequenced using the Illumina platform and PE150 sequencing strategy.

### 2.4. Bioinformatic Analysis

The Illumina high-throughput sequencing results were converted into Raw Reads and stored in FASTQ file format. The low-quality sequences and connector contamination of Raw Reads sequenced from the Illumina platform were removed, the Clean Reads were analyzed. The filter sequencing sequence of each sample was aligned with the reference genome (*Sus scrofa* 11.1) to locate it to the genome by an improved BWT algorithm [16] of HISAT2 software. The number and proportion of genes compared to the genome were called mapping reads and mapping rate. Gene expression level was measured by Fragments per Kilobase per Million Mapped Fragments (FPKM) [17], the most commonly used method for estimating gene expression abundance. DEGSeq [18] was used to analyze DEGs (|Fold Change| > 2 and *p*-value < 0.01). The Principal Component Analysis (PCA), Gene Ontology (GO) term enrichment analysis, Kyoto Encyclopedia of Genes and Genomes (KEGG) pathway enrichment analysis and GSEA were performed using Omicshare, a real-time interactive online data analysis platform (*p*-value < 0.05) (http://www.omicshare.com (accessed on 18 February 2022)) [19].

### 2.5. Real-Time qPCR

Total RNA was extracted from adipose tissue of backfat using TRIzol reagent, and cDNA was synthesized using PCR at 95 °C for 3 min, then 10 s of 40 cycles of 95 °C, 60 °C for 1 min, and 72 °C for 10 s. With TBP as internal control, gene expression levels were measured by real-time quantitative PCR using the 2^−ΔΔCt^ method. The design of primers was developed based on the NCBI Primer-BLAST online tool (https://www.ncbi.nlm.nih.gov/tools/primer-blast/ (accessed on 25 February 2022)). The primers used in this study are displayed in Table 2.

### 2.6. Statistics

All results were analyzed by GraphPad Prism 9, and all data were presented as the means ± standard deviation (SD). The unpaired two-tailed *t*-test was used for statistical analysis. If *p*-value < 0.05 (*), the differences were considered statistically significant.

## 3. Results

### 3.1. Phenotype and Overview of Transcriptomics Data between Duroc and Luchuan Pig Adipose Tissues

It is well known that the Duroc pig is a lean pig breed and the Luchuan pig is a fatty pig breed, partly because the diameter and area of adipocytes in the Luchuan pig are larger than those in the Duroc pig (Figure 1A). In addition, our results show that the backfat of Duroc is thinner than that of Luchuan pigs (Figure 1B). Therefore, there is a positive correlation between adipocytes and fat thickness.

To unveil the mechanism behind the differences in adipose tissue, RNA-seq was conducted to investigate the gene expression profiles at the transcriptional level. The RNA-seq library of Duroc and Luchuan pigs was prepared and sequenced on Illumina platforms, resulting in 45.1–47.9 million and 43.9–46.3 million clean reads for Duroc and Luchuan pigs (Table 3). The identification and quantification information of transcriptome was shown in Appendix A. The PCA of RNA-seq data showed a high correlation between the three biological repeats in Duroc and Luchuan pigs (Figure 1C). The petal Venn diagram shows that 16,162 genes have been identified in all adipose tissue samples (Figure 1D), which had a strong correlation between Duroc and Luchuan pigs (R^2^ = 0.9195; Figure 1E). The abundance distributions of mRNA in Duroc and Luchuan pigs are approximately lognormal (μ = 4; Figure 1F). These results indicate the reliable reproducibility of our data.

### 3.2. Differentially Expressed Genes (DEGs) Identification

Based on RNA-seq data, we can detect the difference in transcriptional mRNA expression in adipose tissue between Duroc and Luchuan pigs. Subsequently, 859 DEGs were screened out (|log_2_ Fold Change| > 1, *p*-value < 0.01), among which 418 genes were highly expressed in the Duroc pig and 441 genes were highly expressed in the Luchuan pig (Figure 2A and Figure 3B)**.** The information of DEGs is shown in Appendix A. The DEGs were further distinguished by supervised hierarchical clustering (Figure 2C). Therefore, we found significantly differentially expressed genes in adipose tissues between Duroc and Luchuan pigs.

### 3.3. GO and KEGG Functional Enrichment Analysis of Duroc and Luchuan Pig Adipose Tissues

GO and KEGG analyses were performed to examine the function of DEGs between Duroc and Luchuan pigs. As shown in Figure 3A, the GO biological processes classified by −log_10_
*p*-value were mainly enriched in processes related to lipid metabolism, concluding positive regulation of lipid localization, cellular lipid catabolic process, lipid catabolic process and positive regulation of lipid storage. Furthermore, the KEGG enrichment analysis displayed the pathways that were mainly concentrated in three KEGG A classes, including metabolism, environmental information processing and organismal systems (Figure 3B, *p*-value < 0.05). Among these pathways, 13 pathways, including fatty acid degradation, alpha-linolenic acid metabolism, linoleic acid metabolism, arachidonic acid metabolism, etc., were related to metabolism; three pathways were involved in environmental information processing and four pathways affected organismal systems. Interestingly, so many essential fatty acids metabolism pathways were significantly enriched.

By standardizing the number of DEGs carried on chromosomes by the length of each chromosome, we found significant enrichment on chromosome 12. Therefore, GO annotation was performed on DEGs on chromosome 12. The results showed that these genes were mainly related to glycine biosynthetic process, phospholipase activator activity, lipase activator activity, etc. (Figure 3C). However, the reason why so many DEGs are enriched on chromosome 12 needs to be further studied.

### 3.4. Gene Set Enrichment Analysis

To further understand the mechanism of fat accumulation difference between Duroc and Luchuan pigs, GSEA-KEGG was executed. The results indicated that, compared with Duroc pigs, the KO01212 fatty acid metabolic pathway decreased in Luchuan pigs (Figure 4A). This may mean that, compared with Duroc pigs, the down-regulation of fatty acid metabolic pathway in Luchuan pigs leads to the accumulation of more adipose. Appendix A and Figure 4B present the gene set and expression abundance of the fatty acid metabolic pathway. To check the reliability of bioinformatics results, the RT-qPCR of genes involved in fatty acid metabolic pathway was performed. According to RNA-seq and RT-qPCR data, compared with Duroc pigs, HSD17B4 and HACD2 were significantly decreased in Luchuan pigs (Figure 4C).

## 4. Discussion

In pig breeding, SCF will affect the meat quality and economic value of pork. SCF is the largest part of fat deposition in animals, in which backfat fat is not only the result of SCF deposition, but also the intuitive embodiment of fat deposition ability. SCF deposition is regulated by multiple genes and pathways. In this study, through the comparison of fat transcriptional maps between lean Duroc pigs and fat Luchuan pigs, it was found that the difference in flavor between Duroc and Luchuan pork may be due to the difference in unsaturated fatty acid metabolism. The difference in fat accumulation was caused by the regulation of the fatty acid metabolism pathway.

Fatty acids, especially polyunsaturated fatty acids (PUFA), are major flavor precursors of meat [20]. Cameron et al. [21] showed that PUFA (C18:2, C20:4 and C22:6, etc.) were positively correlated with flavor of meat. The proportion and amount of saturated fatty acids (SFA) and unsaturated fatty acids (UFA) in food are closely related to human health [22]. In UFA, there is a class of fatty acids called essential fatty acids (EFA) that have protective effects on normal body function and health. EFA includes linoleic acid, linolenic acid and arachidonic acid. Therefore, increasing PUFA content and reducing SFA content of meat has become a tendency for meat quality improvement [23]. Figure 3B showed that the metabolic pathways of three EFA, namely alpha-linolenic acid, linoleic acid and arachidonic acid, were significant different, which may cause the differences in pork flavor and nutrition between Duroc and Luchuan pigs. Compared to Lantang pig (a Chinese fat breed) and Landrace pig (a typical lean breed), Yu et al. found that the PUFA of fatty Lantang pigs was significantly higher than that of lean Landrace pigs. The novel regulatory role of Stearoyl-CoA desaturase (SCD) in PUFA deposition was also identified [24]. In this study, fatty acid desaturase 1(FADS1) and fatty acid desaturase 2(FADS2) were included in the set of genes regulating lipid metabolism that we screened (Figure 4B). We found that the abundance of FADS1 and FADS2 expression in Luchuan pigs was higher than that in Duroc pigs. Previous studies have reported that FADS1 and FADS2 are involved in the biosynthesis of PUFA [24,25]. In conclusion, compared with Duroc pigs, Luchuan pork has more unsaturated fatty acids, and better flavor and nutrition, which may be due to regulation of alpha-linolenic acid, linoleic acid and arachidonic acid metabolism pathways and genes in these pathways.

The imbalance of fat synthesis and consumption is the key factor leading to fat accumulation [26]. Through GSEA, we discovered that the fatty acid metabolism signaling pathway was more active in Duroc pigs than in Luchuan pigs (Figure 4A). The gene set of fatty acid metabolism pathway provided key candidate genes, which are reported to regulate fat deposition (Figure 4B). The 17 β-hydroxysteroid dehydrogenase type 4 (HSD17B4) is a functional enzyme of the fatty acid peroxisomal β-oxidation pathway [27]. HSD17B4 can convert active estrogens and androgens into less active forms and further promoted participation in fatty acid and cholesterol metabolism [28]. Further, 3-hydroxyacyl-CoA dehydratase 2 (HACD2) participates in the production of various the very long-chain fatty acids as precursors of membrane lipids and lipid mediators [29]. Therefore, the high expression of the fatty acid metabolism signal pathway decreased subcutaneous fat accumulation in Duroc pigs, and the inhibition of the fatty acid metabolism signal pathway in Luchuan pigs resulted in increased lipid accumulation.

## 5. Conclusions

In conclusion, this research reveals that the accumulation of SCF in Luchuan pigs is related to inhibition of the fatty acid metabolism signal pathway. In addition, the difference in meat flavor between lean Duroc pigs and fatty Luchuan pigs may be due to the difference in the metabolism of EFA, such as alpha-linolenic acid, linoleic acid and arachidonic acid. These results also help to explain the molecular mechanisms of fat deposition between lean and fat pig breeds.

## Figures and Tables

**Figure 1 animals-12-02258-f001:**
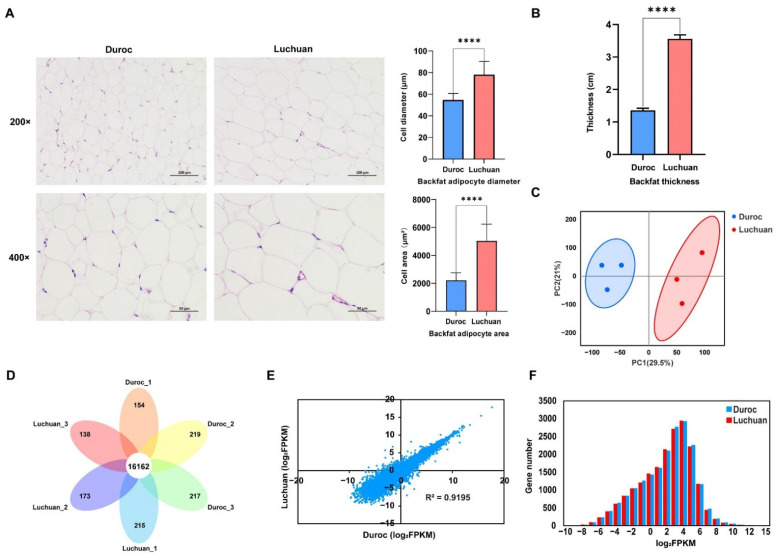
Phenotype and characteristics of RNA-seq data. (**A**) Adipose tissue sections at 200× and 400× magnification. Quantification of the diameter and area of adipocytes. (**B**) Backfat thickness of Duroc and Luchuan. (**C**) PCA of samples. (**D**) The petal Venn diagram of the gene identified by RNA-seq. (**E**) Expression abundance correlation analysis between Duroc and Luchuan in RNA-seq. (**F**) Distribution of the mRNA abundance. The data are expressed as mean ± SD. **** *p* < 0.0001.

**Figure 2 animals-12-02258-f002:**
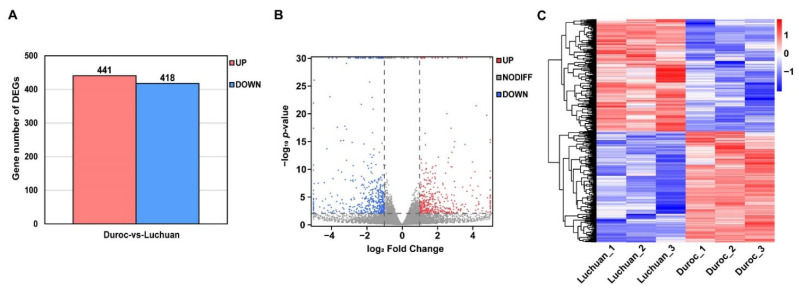
Identification of DEGs. (**A**) The statistical analysis of DEGs between Duroc and Luchuan adipose. (**B**) The volcano map of DEGs. The gene with *p*-value < 0.01 and log_2_ Fold Change > 1 is marked in red; the gene with *p*-value < 0.01 and log_2_ Fold Change < −1 is marked in blue. (**C**) Duroc and Luchuan adipose tissue can be clearly distinguished based on their transcriptome characteristics. The color key (from blue to red) of abundance value indicated low to high expression levels.

**Figure 3 animals-12-02258-f003:**
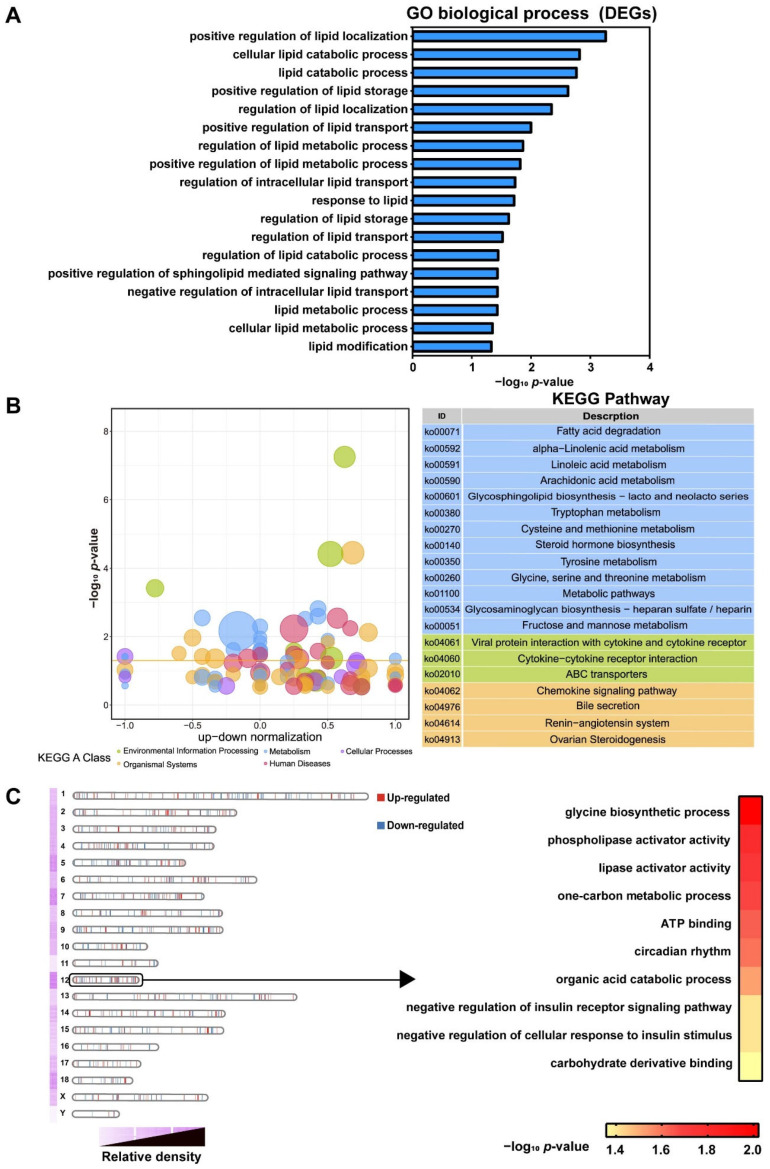
GO and KEGG analysis of DEGs for Duroc and Luchuan adipose. (**A**) GO analysis of DEGs. (**B**) KEGG summary graph showing the summary of the KEGG pathway. Different colors represent different KEGG A class categories. (**C**) Left: distribution of DEGs across chromosomes. Right: significantly enriched GO terms by DEGs in chromosome 12.

**Figure 4 animals-12-02258-f004:**
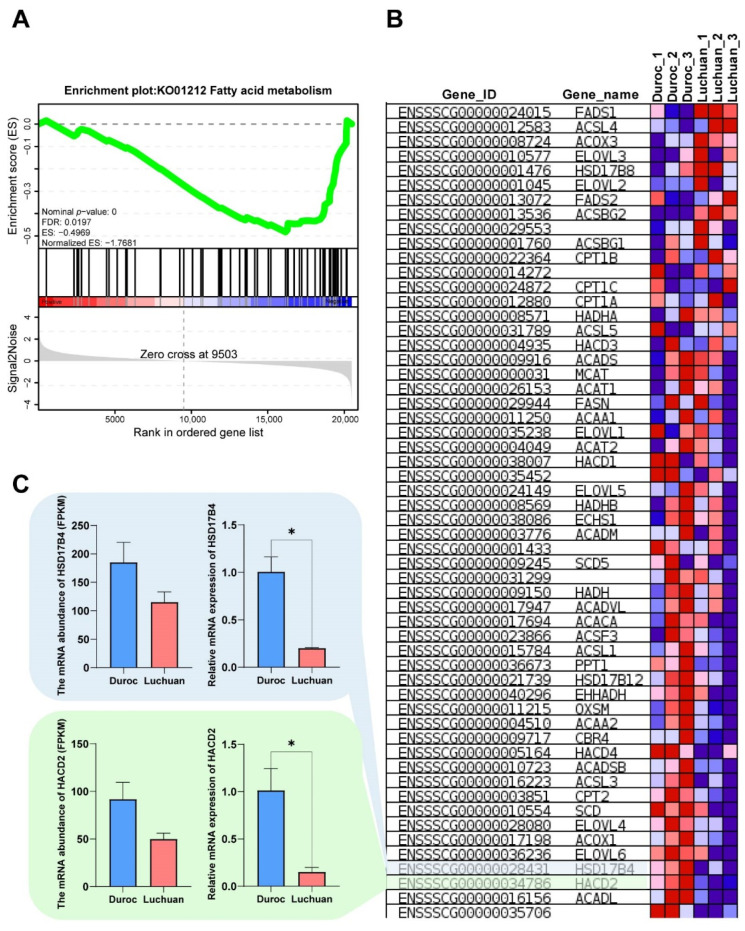
Functional gene screening and functional verification. (**A**) GSEA-KEGG analysis of the transcriptome. A pathway of positive enrichment score is up-regulated, whereas a pathway of negative enrichment score is down-regulated. (**B**) Gene expression heatmap of the fatty acid metabolic pathway. The color key (from blue to red) of abundance value indicated low to high expression levels. (**C**) Gene expression levels quantified by RNA-seq and RT-qPCR. HACD2, 3-hydroxyacyl-CoA dehydratase 2. HSD17B4, hydroxysteroid 17-beta dehydrogenase 4. The data are expressed as mean ± SD. * *p* < 0.05.

**Table 1 animals-12-02258-t001:** The nutritional composition of the diet used in this study.

Nutrition Level	Content (%)
Energy (kj/kg)	11.98
Crude protein (%)	16
Crude fiber (%)	6
Crude ash (%)	7
Calcium (%)	0.6–1.2
Total phosphorus (%)	0.4–1.0
NaCI (%)	0.2–0.8
Lysine (%)	0.8

**Table 2 animals-12-02258-t002:** RT-qPCR primer sequence.

Gene Name	Primer Sequence (5′ to 3′)
TBP_F	GAACTGGCGGAAGTGACGTT
TBP_R	GCACAGCAAGAAAGAGTGATGC
HSD17B4_F	AGGCAGTGGCCAACTATGATTC
HSD17B4_R	AGGAAGAGTTTTCCCCCGATG
HACD2_F	ACTGGAGCCTTGTTGGAGATTT
HACD2_R	ACGCTATGTGTTACTGCCCA

**Table 3 animals-12-02258-t003:** Transcriptome data statistics.

Sample	Raw Reads	Clean Reads	Clean Reads Rate (%)	Q30	Mapped Reads	Mapping Rate
Duroc_1	47,456,140	45,889,484	96.7	95.91%	117,943,789	94.28%
Duroc_2	49,962,578	47,937,230	95.95	95.91%	118,682,360	94.68%
Duroc_3	46,846,584	45,082,868	96.23	95.98%	118,441,470	93.54%
Luchuan_1	45,890,832	43,859,842	95.57	95.90%	115,923,370	91.04%
Luchuan_2	48,171,536	46,262,262	96.04	95.88%	114,285,253	89.87%
Luchuan_3	47,453,720	45,651,590	96.2	95.88%	116,073,783	92.24%

## Data Availability

The results from all data analyzed during this study are included or displayed in this article. However, other individual results are available upon request from the corresponding author.

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
