# Peer review of "Transcriptome Analysis of the Adipose Tissue of Luchuan and Duroc Pigs"

_animals, 2022, doi:10.3390/ani12172258_

Round 1

Reviewer 1 Report (Previous Reviewer 1)

The document has improved significantly, however I still have some comments for the co-authors.

1.- You argue that there are other works where few animals are used for transcriptomic analysis, however, you worked with minimal samples, it is recommended that in future works, you consider samples >3, also, another option is to make an RNA pool of each group of animals.

2. What RIN values ​​of the bioanalyzer did the samples present?

3.- In total you obtained 6 transcriptomes? 3 for each group? And how did you analyze these transcriptomes? How did you filter the data? Please give a brief description of the methodology of how you compared three transcriptomes from the same group?

4.- Line 231: Cameron et al. [20]

5.- In the results, they should include within the text, a table with the values ​​of the quality parameters for the pig breed group, as a total  number of reads, Number total of HQ reads etc.

6.- They should include a table, the top 10 most significantly affected genes. Up/Down-Regulated

5.- In the discussion: In terms of application, which of the two breeds of pigs is better accepted by the consumer? And what is the aim of identifying the genes involved in the taste of meat?

Author Response

Response to Reviewer 1 Comments

Dear reviewer, the parts we have changed in the main text have been highlighted in red. The original figure 1 has been changed to a Graphical abstract, so all the figure numbers have been reduced by 1 (Figure 2-5 had been changed to Figure 1-4).

Comments and Suggestions for Authors

The document has improved significantly, however I still have some comments for the co-authors.

1.- You argue that there are other works where few animals are used for transcriptomic analysis, however, you worked with minimal samples, it is recommended that in future works, you consider samples >3, also, another option is to make an RNA pool of each group of animals.

Response: Your suggestion is very important and I agree with it. In future work, I will use more than 3 biological replicates.

  1. What RIN values of the bioanalyzer did the samples present?

Response: The RIN values of all samples are between 7.8 and 8.1. The specific values are as follows: Duroc_1=8.0; Duroc_2 = 8.0; Duroc_3 = 7.9; Luchuan_1 = 8.1; Luchuan_2 = 7.8; Luchuan_3 = 8.1

3.- In total you obtained 6 transcriptomes? 3 for each group? And how did you analyze these transcriptomes? How did you filter the data? Please give a brief description of the methodology of how you compared three transcriptomes from the same group?

Response: Yes, we obtained a total of 6 transcriptomes, with 3 biological replicates for both the Duroc and Luchuan groups. Analyzing and filtering transcriptome data is described in the Materials and Methods section. For the comparison method of three biological replicates in the same group, we calculated the intra-group similarity by PCA, and the mean expression abundance (FPKM value) of the three biological replicates was the expression abundance of a gene in the group.

4.- Line 231: Cameron et al. [20]

Response: Thank you for pointing this out and we have revised it in the text.

5.- In the results, they should include within the text, a table with the values of the quality parameters for the pig breed group, as a total number of reads, Number total of HQ reads etc.

Response: Thanks for your suggestion. We have added the quality control content in Table 3 of the article.

6.- They should include a table, the top 10 most significantly affected genes. Up/Down-Regulated

Response: According to your suggestion, the information of all differentially expressed genes is shown in Supplementary Table 2. Readers can not only obtain the information of top10 up-regulated and down-regulated genes from supplementary table 2, but also obtain the information of all differential expressed genes, including the read count, fold change, significance value, up-regulated and down-regulated genes, gene function annotation and other information.

7.- In the discussion: In terms of application, which of the two breeds of pigs is better accepted by the consumer? And what is the aim of identifying the genes involved in the taste of meat?

Response: In terms of application, consumers prefer pork with high lean meat percentage, rich intramuscular fat and better pork flavor. However, producers prefer to feed pigs that grow faster and have a low feed to meat ratio. Duroc pig is famous for its thin backfat, high rate of lean meat and fast growth. Luchuan pig has thick backfat, high fat content, slow growth and better pork flavor. Both of them have advantages and disadvantages. Therefore, by comparing Duroc pigs and Luchuan pigs, we aim to obtain valuable information to guide production. The identification of genes related to the regulation of meat flavor can provide reference and target for breeding pig breeds with better flavor.

Reviewer 2 Report (Previous Reviewer 3)

General Comments (For Author)

This paper is considered to be an interesting paper on RNA-seq analysis for adipocyte of Luchuan and Duroc Pig. In this study, the purpose of the study is very impressive and I think that important data have been obtained. However, I think there is something lacking, in explaining the results of the analysis.

Major Comments

1. (Introduction) I wonder why the authors compared Luchuan and Duroc Pig. Especially since there are many commercial species, is there any particular reason the author used Duroc as a comparison?

2. (Introduction) Why did author think the expression of adipocytes was related to fat thickness in mature pig?

3. (Result) As shown in Figure 1, GSEA analysis was not performed with DEG. With which genes were the GSEA analysis performed?

4. (Result) In this study, the difference in the diameter and area of ​​adipocytes was shown. It is necessary to explain how these characteristics are related to the thickness of fat.

5. (Discussion, Line 229) The author described fatty acids and flavors, but I thought this part was out of the question of looking at the relationship between fat thickness and gene expression. Also, since there were no additional data related to flavor or gene expression results, I thought it should be explained how the transcriptome analysis results are related to fat thickness in pig.

6. Overall, the results of gene function analysis ware related to the lipid or fat mechanism, but it is necessary to clearly explain which genes and which gene groups are related to the fat thickness.

Author Response

Response to Reviewer 2 Comments

Dear reviewer, the parts we have changed in the main text have been highlighted in red. The original figure 1 has been changed to a Graphical abstract, so all the figure numbers have been reduced by 1 (Figure 2-5 had been changed to Figure 1-4).

Comments and Suggestions for Authors

General Comments (For Author)

This paper is considered to be an interesting paper on RNA-seq analysis for adipocyte of Luchuan and Duroc Pig. In this study, the purpose of the study is very impressive and I think that important data have been obtained. However, I think there is something lacking, in explaining the results of the analysis.

Major Comments

  1. (Introduction) I wonder why the authors compared Luchuan and Duroc Pig. Especially since there are many commercial species, is there any particular reason the author used Duroc as a comparison?

Response: Thank you for your comment. Duroc is a typical lean pig breed with less subcutaneous fat deposition and a backfat thickness of 1-2 cm. Luchuan pig is a typical fatty pig breed, with more subcutaneous fat deposition and backfat thickness of 3-4 cm. We compared the adipose tissue of these two pig breeds to highlight the differences in their ability to deposit fat. We used these two pig breeds to compare to find key genes involved in fat development and deposition. In addition, Duroc pigs and Luchuan pigs were selected for this study because no adipose tissue transcriptome studies of these two pigs have been reported so far. Another reason is that subsequent research projects need to be based on the transcription profiles of Duroc and Luchuan pigs.

  1. (Introduction) Why did author think the expression of adipocytes was related to fat thickness in mature pig?

Response: Thank you for your comment. Fat deposition is characterized by an increase in the number and size of adipocytes [1]. Our results showed that fatty Luchuan pigs had thicker backfat thickness (Figure 1B) and larger adipocyte area and diameter (Figure 1A) than lean Duroc pigs. Therefore, we believe that there is a positive correlation between adipocytes and fat thickness.

  1. (Result) As shown in Figure 1, GSEA analysis was not performed with DEGs. With which genes were the GSEA analysis performed?

Response: Thank you for such a detailed question. GSEA is a genome-wide analysis using the expression levels of all genes. We have supplemented the gene range analyzed by GSEA in Graphical abstract. GSEA can avoid the situation that some genes with important biological functions but not obvious expression differences are ignored due to the artificially set threshold of screening differentially expressed genes. In addition, compared with GO and KEGG analysis, GSEA can obtain up-down-regulation information of pathway.

  1. (Result) In this study, the difference in the diameter and area of adipocytes was shown. It is necessary to explain how these characteristics are related to the thickness of fat.

Response: Thank you for your suggestion. Fat deposition is characterized by an increase in the number and size of adipocytes[1]. Our results showed that fatty Luchuan pigs had thicker backfat thickness (Figure 1B) and larger adipocyte area and diameter (Figure 1A) than lean Duroc pigs. Therefore, we believe that there is a positive correlation between adipocytes and fat thickness.

  1. (Discussion, Line 229) The author described fatty acids and flavors, but I thought this part was out of the question of looking at the relationship between fat thickness and gene expression. Also, since there were no additional data related to flavor or gene expression results, I thought it should be explained how the transcriptome analysis results are related to fat thickness in pig.

Response: Thank you for pointing out this profound problem. Indeed, the original intention of this paper was to explore the mechanism of fat accumulation in pigs by transcriptome sequencing of back adipose tissue. However, we found that fatty acids were related to the flavor of meat during the study, so we discussed them in the hope of promoting the research on the flavor of meat. Most importantly, the most vital result of this paper is also clearly discussed. The difference in fat accumulation between Duroc and Luchuan pigs is the differential expression of fatty acid metabolism signal pathway, and the differential gene expression profile is provided.

  1. Overall, the results of gene function analysis ware related to the lipid or fat mechanism, but it is necessary to clearly explain which genes and which gene groups are related to the fat thickness.

Response: Thank you for your valuable advice. We added the gene set of fatty acid metabolism signaling pathway concluded in this paper in Supplemental Table 3.

[1] J. Shao, X. Bai, T. Pan, Y. Li, X. Jia, J. Wang, S. Lai, Genome-Wide DNA Methylation Changes of Perirenal Adipose Tissue in Rabbits Fed a High-Fat Diet, Animals: an open access journal from MDPI, 10 (2020).10.3390/ani10122213

Round 2

Reviewer 2 Report (Previous Reviewer 3)

The authors did their best to answer all the questions.

This manuscript is a resubmission of an earlier submission. The following is a list of the peer review reports and author responses from that submission.

Round 1

Reviewer 1 Report

It is an interesting study, however it has problems in the methodology, it is necessary to detail more how the samples were obtained, Why did they only use three samples?. In the RNAreq analysis they used an RNA pool or adipose tissue?, in the bioinformatic analysis section, the values were not indicated of significance for DEGS and KEGG; the discussion is poor, it is necessary to better discuss the results with other studies where the metabolic pathways of the fatty acids reported in this study are involved.

Citations and references do not follow editorial instructions.

Line: 
16: both races, correspond to Duruc?
19: 859 differentially expressed genes, how many for each breed?.
29: Change the word "ideas" to the word "hypothesis".
31: include the word "lipid metabolism" in kerwords.
45-48: They argue that the local Luchuan pig has a greater capacity for fat deposition and a low percentage of lean meat. They have to argue better because they need to know the genes that are expressed in each race. How useful is it to know the genes that are expressed in the Luchuan race? Is there a way that it can be silenced or induced, so that in the future the proportion of lean meat can be increased and fat reduced?
50-55: This paragraph is out of context, or it lacks key words that allow linking the idea, with the text of lines 56-58.
97: It remains to be explained, how the samples of each race were analyzed, was the transcriptome analyzed individually? Or was a pool of tissue or RNA made and subsequently sequenced?
98: What value does the NanoPhotometer® spectrophotometer use to determine that the RNA is of good quality? I need to include it
98: In the bioinformatic analysis section, it is necessary to describe better and with more data. for example: how many raw reads were assembled ? lack analysis quality reads, lacks  p-value assigned to each gene to evaluate its statistical significance. In the analysis of DEGs, it is missing to include, multiple-testing corrections, using the Benjamini and Hochberg step-up false-discovery rate (FDR)-controlling procedure to calculated adjusted p-values.
131: It is important to indicate the design of the primers and the genes that were used for the validation of the differentially expressed genes.
150: 285,781,390 raw reads and 274,683,276 clean reads,Is it for each library or for both? Please be more specific.
181: A classes?
220: In the discussion, it was not possible to relate the results obtained with the importance of the Luchuan breed, why is it important to improve the breed? Does the meat taste better than the Duruc breed? What does the consumer require? flavor of meat, which is provided by lipids, or lean meat without flavor?.
225: Can you explain how you can improve the genetic base with the information generated from the DEGs?
230: as is the behavior of the metabolism of the amino acids serine, threonine and cysteine, and the metabolism of fructose and mannose in the Luchuan vs. Duroc breed, which suggests that due to this behavior, the Luchuan breed deposits a greater amount of fat. Explain in the text.
235-237: It is necessary to better support this statement, the flavor of the meat is due to the different fatty acids. include more citations.

Reviewer 2 Report

 To Authors

The proposed research presented in the manuscript concern on explore the key genes regulating fat deposition used pigs of two breeds and analyzing transcriptome sequencing. In my opinion the selected section is not correct, more appropriate would be the animal genetics. 

Below the details of reviewed work:

  1. The chapter Introduction:

In the first part of that chapter Authors described pigs breeds used in research. In the second part they characterized pigs like the ideal animal models for studying human diseases. What for? Next they described transcriptome sequencing. This chapter is not connected with Animal nutrition section.  Besides, the aim of the study is not clear and there is no research hypothesis.

  1. The chapter Materials and Methods:

The lack of information about animals:

  • Animal housing system,
  • Animal management (Directive No. 2010/63/EU),
  • Animal nutrition - the composition of feed (dietary value, composition of the fodder),
  • Ethics Approval?
  • Animals were euthanized, why? Not slaughtered in the slaughterhouse?
  • Production traits results?

Besides, Authors used only 6 animals in experiment (3 Duroc pigs breed and 3 Luchuan breed pigs). These breeds are different in aspect of growth rate and fatness and meatiness of carcasses. Luchuan breed is a one of local pig breed in China and characterized by high marbling of the meat than meat obtained from Duroc breed pigs. So, in that case it should be indicated not only the age of fatteners but also body weight on the end of experiment.

  1. The chapter Results:

There is no results about production traits of pigs: fattening (growth rate, daily weight gain, feed consumption per 1 kg of weight gain), slaughter value (carcass yield, meatiness, backfat thickness), and meat quality.

  1. The chapter Conclusions:

Authors in conclusion chapter indicate that presented results provided a reference for the future improvement of pork quality. What kind reference? 

In my opinion the topic of that research is not connected with Animal Nutrition. So, manuscript should be send to animal genetics journal/section. The work isn’t suitable for publication in the Animals journal.

Reviewer 3 Report

General Comments (For Author)

This paper is considered to be an important paper on identification of transcriptomic difference between adipocytes of two breeds. This study was well explained and the results were well organized. However, before making the decision to accept this paper, I think the author needs to answer some simple questions.

Major Comments

  1. (Introduction) I think it is necessary to clarify why the author compared the Chinese native specie (Luchuan) to Duroc for adipocytes in this study. What do author think is the biggest difference between the adipocytes of two breeds?
  2. (Method) The author used three samples per each breed to compare the transcriptomes. It is necessary to describe the rationale that these samples are representative. Also, the transcriptome is not always the same at each stage of growth in pigs, so why was it sampled at 180 days of age? This is the period after the fat cells have all grown. Is it a time for sampling suitable for the purpose of the study?
  3. (Result) Even though the transcript values ​​for all genes were compared, why didn't multiple tests be performed? (In this study, p-value of 0.01 was used)
  4. (Result) What is the biological significance of selecting meaningful chromosomes?
  5. (Discussion) The interpretation of the DEG discovered in this study is insufficient.

It is important to show the results of multiple analyzes, but it is also important to clearly state which differences the author attaches important meaning to in the results of the analysis.

Minor Comments

  1. In order to provide information about the relationship between samples, please draw a PCA using 6 sample data.
  2. If the text in Figure 3 is slightly enlarged, it seems that the readability will be improved.